# Single crystal growth of PrNiGe$_2$ compounds

**Kousei Ishiwatari[1], Kaito Hoshii[1], Keisuke Ida[1] and Masashi Ohashi[2]⋆**

**1** Graduate School of Natural Science and Technology, Kanazawa University,
Kakuma-machi, Kanazawa 920-1192, Japan
**2** Institute of Science and Engineering, Kanazawa University,
Kakuma-machi, Kanazawa 920-1192, Japan

⋆ ohashi@se.kanazawa-u.ac.jp

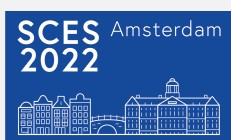 *International Conference on Strongly Correlated Electron Systems
(SCES 2022)
Amsterdam, 24-29 July 2022*

## Abstract

We report on the magnetic characterization of a praseodym intermetallic PrNiGe$_2$. Several single crystals are grown by the Czochralski method from Ni- deficient sample as the initial one. X-ray analysis of PrNi$_{0.8}$Ge$_2$ indicated the CeNiSi$_2$-type structure as the only phase. The magnetic measurements clearly indicate that PrNi$_x$Ge$_2$ exhibits a ferromagnetic orderings at 12 K, which is independent with of composition of Ni. A strong anisotropy along the three principal crystallographic directions was observed, reflecting the orthorhombic symmetry of the crystal structure. The $b$-axis was found to be the easy axis of magnetization.

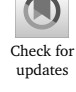
## 1 Introduction

The ternary compounds CeTX$_2$ (T = transition metal and X = Si, Ge, Sn) form a large family having the orthorhombic CeNiSi$_2$-type layered structure, which is constructed from deformed fragments of the CeGa$_2$Al$_2$ and $\alpha$-ThSi$_2$ structures [1–3]. The lattice parameter along $b$-axis is extremely large compared to those along $a$- and $c$- axes, and it is expected that highly anisotropic magnetic property exists. Indeed, these compounds have received considerable interest of a great variety of magnetic behaviors [4–6]. There are not so many reports on single crystal growth due to the difficulty in obtaining single phase samples. On the other hand, the magnetic properties of polycrystals of RT$_x$Ge$_2$ (T: Fe, Co, Ni, Cu, $x < 1$) have been investigated in a previous study, and it seems to be easier to grow polycrystals when the amount of $T$ is reduced [7–10]. PrNiGe$_2$ is known to order ferromagnetically below 13 K [11]. In this study, we have grown a single crystal of PrNi$_x$Ge$_2$ and report the anisotropic magnetic properties here.

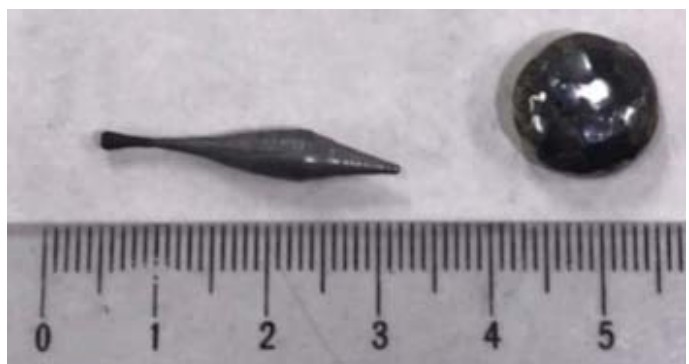

Figure 1: Photograph of a single crystal ingot of $PrNi_{0.8}Ge_2$.

## 2 Experimental

The polycrystalline samples of $PrNi_xGe_2$ ($x = 0.6$, 0.8, and 1.0) were prepared as starting materials by arc-melting praseodym (99.9% ), nickel (99.99% ) and germanium (99.999% ) in a tetra-arc furnace under argon atmosphere. To improve homogenity, the ingot was turned over and remelted several times. The weight loss is negligible. Single crystals were grown by a Czochralski pulling method. The pulling parameter was kept constant during the growth (pulling rate: 10 mm/h; seed rotation speed: 10 rpm; crucible-rotation speed: 5 rpm). An ingot was 2-3 mm in diameter and 30 mm in length as shown in Figure 1. Its homogeneity and chemical composition were checked by microprobe analysis, which was made using JEOL SEM and Oxford Instrument EDX at Venture Business Laboratory Kanazawa University, based on the measurement of the Ce $L_{\alpha 1}$, Ni $K_{\alpha 1}$ and Ge $K_{\alpha 1}$ X-ray emission lines. The experimental atomic percentages of $PrNi_xGe_2$ are obtained to be Pr 25.3%, Ni 23.7% and Ge 50.9% for $x = 1.0$, Pr 30.0%, Ni 26.6% and Ge 43.3% for $x = 0.8$, and Pr 21.5%, Ni 16.4% and Ge 61.9% for $x = 0.6$, respectively.

The samples were checked by conventional x-ray powder diffraction experiments using Cu-$K_\alpha$ radiation. The single crystalline state was confirmed using back-scattering Laue technique. The dc magnetic susceptibility was measured in the temperature range 2.0-300 K using a Quantum Design MPMS-5 superconducting quantum interference device magnetometer. The specific heat was measured by utilizing the Heat Capacity option on a Physical Properties Measurement System.

## 3 Results and Discussion

Figure 2 shows X-ray powder diffraction pattern of $PrNi_xGe_2$. All diffraction peaks can be indexed in the orthorhombic $CeNiSi_2$ type layered structure (space group Cmcm), as shown in the calculated spectrum. The lattice parameters $a, b$, and $c$ are determined by a least-squares approximation from the $d$-values corresponding to the reflections, and are gathered in Table 1. They are consistent with those of previous reports [7, 11].

The specific heat of $PrNi_xGe_2$ single crystal is shown in Figure 3. All samples show an anomaly reaching a maximum of $\sim$14 J mol$^{-1}$K$^{-1}$ at 12 K. It is consistent with the previous result of $PrNiGe_2$ polycrystalline sample [11], which is associated with the onset of ferromagnetic order. A small anomaly is observed only in $PrNi_{0.6}Ge_2$ at $\sim$17 K, indicating some magnetic impurity may be included because the sample deviate from stoichiometry.

Figure 4 shows the temperature dependence of $M/H$ in a magnetic field of 1 kOe and the

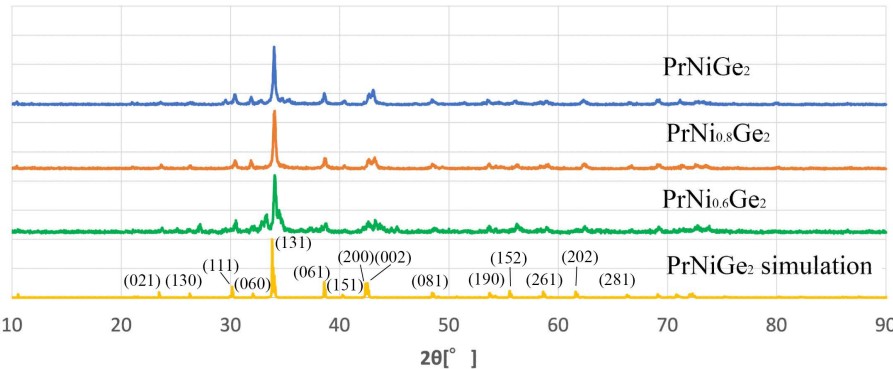

Figure 2: The X-ray powder diffraction pattern of $PrNi_xGe_2$. Indices are shown for the peaks of the calculated spectrum.

Table 1: Lattice parameters in the orthorhombic structure of XRD patterns and the unit cell volume of $PrNi_xGe_2$.

| $x$ | $a$ (Å) | $b$ (Å) | $c$ (Å) | $V$ (Å$^3$) |
|-----|---------|---------|---------|-------------|
| 1.0 | 4.235 | 16.80 | 4.212 | 299.7 |
| 0.8 | 4.237 | 16.78 | 4.191 | 298.0 |
| 0.6 | 4.267 | 16.77 | 4.201 | 300.6 |

inverse magnetic susceptibility of $PrNi_xGe_2$. A strong decrease in magnetization is observed at around 13 K for both $x=1.0$ and 0.8 samples, which is consistent with the result of specific heat measurement. The magnetic susceptibilities of those compounds in the paramagnetic state are fitted using the following formula:

$$M/H(T) = \frac{N\mu_{\text{eff}}^2}{3k_B(T-\Theta)} + \chi_0,\tag{1}$$

where $N$ is the Avogadro number, $k_B$ is Boltzmann's constant, $\Theta$ is the paramagnetic Curie temperature, and $\chi_0$ is a temperature-independent part of the magnetism including the diamagnetic core correction, the Pauli susceptibility of the electron gas, or the Van Vleck temperature independent paramagnetism. Fitting was carried out in the temperature range $100\ \text{K} < T < 180\ \text{K}$. $\Theta$, $\chi_0$, and the effective magnetic moment $\mu_{\text{eff}}$ are given in Table 2. The $\chi_0$ values are relatively large for both $x=1.0$ and 0.8 compounds while the $\mu_{\text{eff}}$ values are in good agreement with the Hund's rule ground state for $Pr^{3+}$ ion (3.58 $\mu_B$).

Table 2: Magnetic data for $PrNi_xGe_2$ compounds.

| | $PrNiGe_2$ $b$-axis | $PrNi_{0.8}Ge_2$ $a$-axis | $PrNi_{0.8}Ge_2$ $b$-axis | $PrNi_{0.8}Ge_2$ $c$-axis |
|---|---|---|---|---|
| $\Theta$ (K) | 46.1 | $-6.1$ | 48 | $-6.0$ |
| $\mu_{rmeff}$ ($\mu_B$) | 3.47 | 3.06 | 3.34 | 2.99 |
| $\chi_0$ ($10^{-3}$emu/mol) | 1.93 | 1.63 | 1.88 | 1.78 |
| $M[5T, 5K]$ ($\mu_B$) | 2.55 | 0.679 | 2.89 | 0.982 |

The isothermal magnetization of $PrNi_xGe_2$ measured at $T = 5.0$ K along the three principal crystallographic directions is shown in figure 5 (a). For $H//b$-axis the magnetization increases

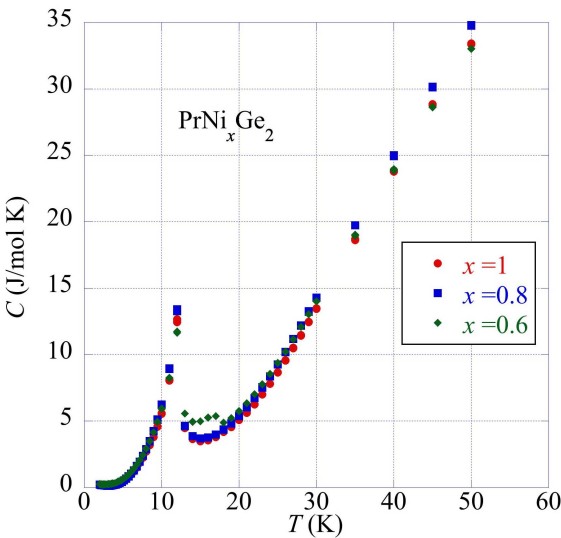

Figure 3: The specific heat of PrNi$_x$Ge$_2$.

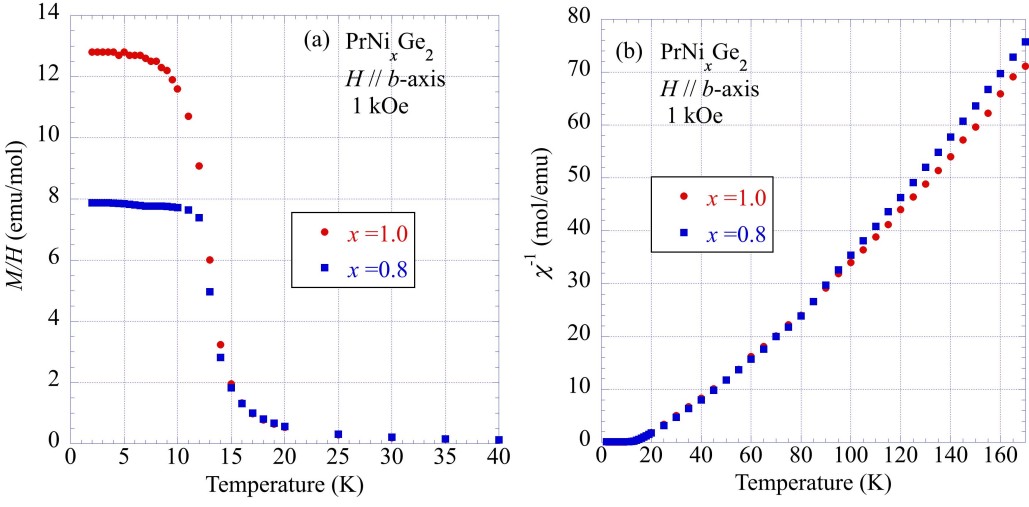

Figure 4: (a) $M/H$ and (b) the inverse magnetic susceptibility as a function of temperature in a magnetic field of 1 kOe along $b$−axis of PrNi$_x$Ge$_2$.

more rapidly with field than along the other two directions, thus indicating the $b$-axis as the easy axis of magnetization. Hysteretic behaviour is observed especially along $b$-axis, confirming the ferromagnetic ground state. At 5.0 K the magnetization saturates to 2.55 $\mu_B$/Pr and 2.89 $\mu_B$/Pr for PrNiGe$_2$ and PrNi$_{0.8}$Ge$_2$, respectively, which is larger than that of a PrNiGe$_2$ porycrystalline sample [12]. The saturation moment of the free Pr$^{3+}$ ion ($g_J J \mu_B = 3.20\mu_B$) is relatively close to the saturation magnetization of PrNi$_{0.8}$Ge$_2$.

Figure 5 (b) show the temperature dependences of the inverse magnetic susceptibility of PrNi$_{0.8}$Ge$_2$. The anisotropy in the magnetic susceptibility along the three principal directions is clearly evident. The magnetic susceptibilities in the paramagnetic state are fitted using the formula (1). Fitting was carried out in the temperature range 100 K $< T <$ 180 K. The Weiss temperature Θ is positive for $b$-axis, as expected for a ferromagnetic ordering compound. On the other hand, Θ is negative for $a$- and $c$-axes.

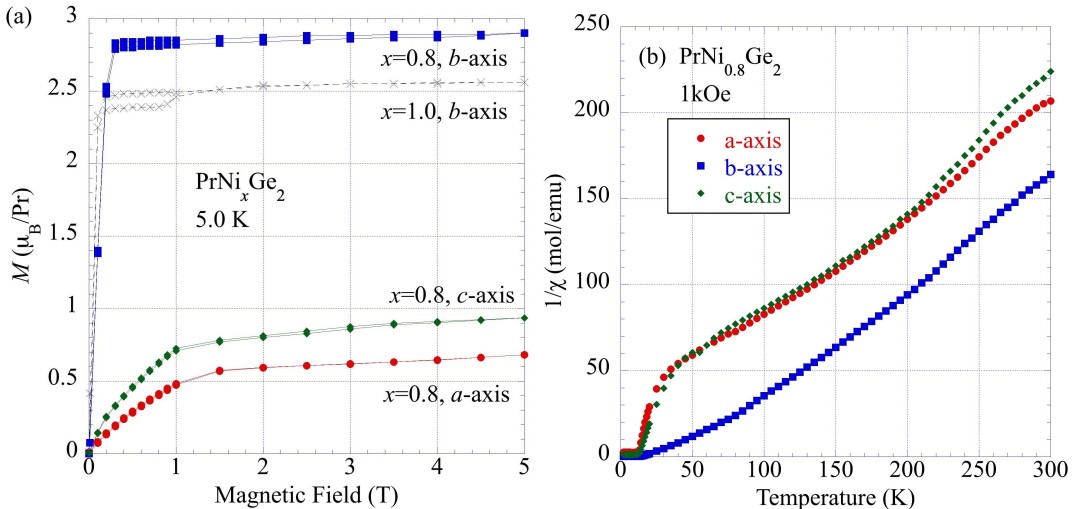

Figure 5: (a) The inverse magnetic susceptibility as a function of temperature (b) The field dependence of the magnetization of $PrNi_{0.8}Ge_2$ at 5.0 K in a magnetic field of 1 along $a$-, $b$-, and $c$-axes of $PrNi_{0.8}Ge_2$.

## 4 Summary

In this study $PrNiGe_2$ single crystals are grown by the Czochralski method from Ni- deficient sample as the initial one. X-ray analysis of the sample indicated the $CeNiSi_2$-type structure as the only phase. The specific heat and the magnetic measurements clearly indicate that $PrNi_xGe_2$ exhibits a ferromagnetic orderings at 12 K, which is independent with of composition of Ni. A strong anisotropy along the three principal crystallographic directions was observed, reflecting the orthorhombic symmetry of the crystal structure. The $b$-axis was found to be the easy axis of magnetization. From the result of the magnetic susceptibility along the $b$- plane, the effective magnetic moment is close to the value expected for the free $Pr^{3+}$ ions, while a temperature-independent part of the magnetism remains.

## Acknowledgements

This work was financially supported by Takahashi industrial and economic research foundation, Shibuya science culture and sports foundation, and Yamada science foundation.

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
