# Peer review of "Single crystal growth of PrNiGe2 compounds"

_SciPost Physics Proceedings, doi:SciPost Phys. Proc. 11, 002 (2023)_

## Round 1 · Referee Report · Anonymous (Referee 1) · 2022-10-28

Report

The manuscript reports the growth of PrNi1-xGe2 single crystals using the Czochralski method and characterisations of the crystals through powder X-ray diffraction and heat capacity measurements. The manuscript also reports strong magnetic anisotropy in PrNi1-xGe2 observed through DC magnetisation measurements. Although lacking a discussion of the underlying physics behind the observed strong anisotropy, the information included in the manuscript will be useful for those who are interested in this and related materials, and it is worth publishing it in a conference proceeding. However, I would like to request several changes to the manuscript before I can recommend its publication. These are listed in the “Requested changes” section.

Requested changes

1 - In the Abstract, the authors state that “X-ray analysis of the sample indicated the CeNiSi2-type structure as the only phase”. This is clearly not true given the unidentified peaks in the X-ray powder diffraction pattern shown in Figure 2 which presumably come from secondary phases. Please rewrite this sentence according to the facts.

2 - There appears to be an extra peak with significant weight around 2θ = 43 degrees in all three measured X-ray patterns, which is not seen in the simulated pattern. Can the authors identify possible candidates of these secondary phases?

3 - The first paragraph of the Introduction contains a generic description of Kondo lattice physics, which seems irrelevant to the rest of the manuscript. So, I recommend the authors to simply remove those sentences from the introduction. Instead, I recommend the authors to expand the introduction on the reported magnetic properties in RTxGe2, and how that compares with PrNixGe2.

4 - Throughout the manuscript, the ferromagnetic ordering temperature stated by the authors has been varying, for instance, 14 K in the abstract but 12 K in the conclusion. Please specify a clear criterion, based on which the ordering temperature has been determined, and keep the value consistent throughout the manuscript.

5 - In Table 1, the lattice parameters for x = 0.8 and x = 0.6 are identical. I believe this is a typo. Please correct it.

6- Please specify how the lattice parameters are determined from the powder X-ray diffraction patter, for example by Rietveld refinement using which software.

7 - The EDX result reported by in Section 2 for the three samples appear to vary strongly between the three samples. In particular, the measured composition for x = 0.8 seems to be significantly deficient in Ge. Can the author quantify the error on these composition measurements or provide an explanation as to the strong variation in the measured compositions?

8 - For the fitting of magnetisation data to Equation (1), please specify the temperature window for which the fitting has been carried out in each case.

  • validity: good
  • significance: low
  • originality: ok
  • clarity: good
  • formatting: excellent
  • grammar: good

Author:  Masashi Ohashi  on 2022-12-06  [id 3104]

(in reply to Report 1 on 2022-10-28)

First, we would like to thank the Referee for taking their time to read and comment on our lengthy manuscript. We also greatly appreciate valuable and helpful questions and suggestions for improving our manuscript. We address all of them point by point below. We have worked hard to incorporate your feedback and hope that these revisions persuade you to accept our submission.

On behalf of the authors, Masashi Ohashi

1 - In the Abstract, the authors state that “X-ray analysis of the sample indicated the CeNiSi2-type structure as the only phase”. This is clearly not true given the unidentified peaks in the X-ray powder diffraction pattern shown in Figure 2 which presumably come from secondary phases. Please rewrite this sentence according to the facts. 2 - There appears to be an extra peak with significant weight around 2θ = 43 degrees in all three measured X-ray patterns, which is not seen in the simulated pattern. Can the authors identify possible candidates of these secondary phases?

Response) At least XRD of PrNi0.8Ge2 indicated the CeNiSi2-type structure as the only phase. Compared to the simulation, the peak around 2θ=43 corresponds to the (002) reflection. To make it clear, we replaced "X-ray analysis of the sample" with "X-ray analysis of PrNi0.8Ge2" in the abstract. Indices are added in the XRD simulation in Fig. 1.

3 - The first paragraph of the Introduction contains a generic description of Kondo lattice physics, which seems irrelevant to the rest of the manuscript. So, I recommend the authors to simply remove those sentences from the introduction. Instead, I recommend the authors to expand the introduction on the reported magnetic properties in RTxGe2, and how that compares with PrNixGe2.

Response) We have deleted the first paragraph of the Introduction.

4 - Throughout the manuscript, the ferromagnetic ordering temperature stated by the authors has been varying, for instance, 14 K in the abstract but 12 K in the conclusion. Please specify a clear criterion, based on which the ordering temperature has been determined, and keep the value consistent throughout the manuscript.

Response) Tc is estimated from the result of heat capacity. So, we replaced 14 K with 12 K in abstract.

5 - In Table 1, the lattice parameters for x = 0.8 and x = 0.6 are identical. I believe this is a typo. Please correct it.

Response) We thank for pointing out our serious mistakes. We carefully recalculated and revised Table 1. We found that the unit cell volume does not decrease as decreasing x. So, the relevant sentence in the abstract and the first paragraph in section 3 has been deleted.

6- Please specify how the lattice parameters are determined from the powder X-ray diffraction patter, for example by Rietveld refinement using which software.

Response) The lattice parameters a, b, and c are determined by a least-squares approximation from the d-values corresponding to the reflections. We have added an explanation to the first paragraph in section 3. Indices are added in the XRD simulation in Fig. 1.

7 - The EDX result reported by in Section 2 for the three samples appear to vary strongly between the three samples. In particular, the measured composition for x = 0.8 seems to be significantly deficient in Ge. Can the author quantify the error on these composition measurements or provide an explanation as to the strong variation in the measured compositions?

Response) The composition obtained is an experimental fact. There may be problems with the experimental setup, but we cannot quantify the error. So, we deleted the sentence "These results are closed to the theoretical value..."

8 - For the fitting of magnetisation data to Equation (1), please specify the temperature window for which the fitting has been carried out in each case.

Response) Fitting was carried out in the temperature range 100 K < T < 180 K. We have added a sentence on page 3, line 4 from the bottom and on page 5, line 12.

Attachment:

prnige2_2_ZBU3GM0.pdf

Author:  Masashi Ohashi  on 2022-12-06  [id 3103]

(in reply to Report 1 on 2022-10-28)

First, we would like to thank the Referee for taking their time to read and comment on our lengthy manuscript. We also greatly appreciate valuable and helpful questions and suggestions for improving our manuscript. We address all of them point by point below. We have worked hard to incorporate your feedback and hope that these revisions persuade you to accept our submission.

On behalf of the authors, Masashi Ohashi

1 - In the Abstract, the authors state that “X-ray analysis of the sample indicated the CeNiSi2-type structure as the only phase”. This is clearly not true given the unidentified peaks in the X-ray powder diffraction pattern shown in Figure 2 which presumably come from secondary phases. Please rewrite this sentence according to the facts. 2 - There appears to be an extra peak with significant weight around 2θ = 43 degrees in all three measured X-ray patterns, which is not seen in the simulated pattern. Can the authors identify possible candidates of these secondary phases?

Response) At least XRD of PrNi0.8Ge2 indicated the CeNiSi2-type structure as the only phase. Compared to the simulation, the peak around 2θ=43 corresponds to the (002) reflection. To make it clear, we replaced "X-ray analysis of the sample" with "X-ray analysis of PrNi0.8Ge2" in the abstract. Indices are added in the XRD simulation in Fig. 1.

3 - The first paragraph of the Introduction contains a generic description of Kondo lattice physics, which seems irrelevant to the rest of the manuscript. So, I recommend the authors to simply remove those sentences from the introduction. Instead, I recommend the authors to expand the introduction on the reported magnetic properties in RTxGe2, and how that compares with PrNixGe2.

Response) We have deleted the first paragraph of the Introduction.

4 - Throughout the manuscript, the ferromagnetic ordering temperature stated by the authors has been varying, for instance, 14 K in the abstract but 12 K in the conclusion. Please specify a clear criterion, based on which the ordering temperature has been determined, and keep the value consistent throughout the manuscript.

Response) Tc is estimated from the result of heat capacity. So, we replaced 14 K with 12 K in abstract.

5 - In Table 1, the lattice parameters for x = 0.8 and x = 0.6 are identical. I believe this is a typo. Please correct it.

Response) We thank for pointing out our serious mistakes. We carefully recalculated and revised Table 1. We found that the unit cell volume does not decrease as decreasing x. So, the relevant sentence in the abstract and the first paragraph in section 3 has been deleted.

6- Please specify how the lattice parameters are determined from the powder X-ray diffraction patter, for example by Rietveld refinement using which software.

Response) The lattice parameters a, b, and c are determined by a least-squares approximation from the d-values corresponding to the reflections. We have added an explanation to the first paragraph in section 3. Indices are added in the XRD simulation in Fig. 1.

7 - The EDX result reported by in Section 2 for the three samples appear to vary strongly between the three samples. In particular, the measured composition for x = 0.8 seems to be significantly deficient in Ge. Can the author quantify the error on these composition measurements or provide an explanation as to the strong variation in the measured compositions?

Response) The composition obtained is an experimental fact. There may be problems with the experimental setup, but we cannot quantify the error. So, we deleted the sentence "These results are closed to the theoretical value..."

8 - For the fitting of magnetisation data to Equation (1), please specify the temperature window for which the fitting has been carried out in each case.

Response) Fitting was carried out in the temperature range 100 K < T < 180 K. We have added a sentence on page 3, line 4 from the bottom and on page 5, line 12.

Attachment:

prnige2_2.pdf

---

## Round 2 · Author Response

First, we would like to thank the Referee for taking their time to read and comment on our lengthy manuscript. We also greatly appreciate valuable and helpful questions and suggestions for improving our manuscript. We address all of them point by point below. We have worked hard to incorporate your feedback and hope that these revisions persuade you to accept our submission.

On behalf of the authors,
Masashi Ohashi

---

## Round 2 · List of Changes

1 - In the Abstract, the authors state that “X-ray analysis of the sample indicated the CeNiSi2-type structure as the only phase”. This is clearly not true given the unidentified peaks in the X-ray powder diffraction pattern shown in Figure 2 which presumably come from secondary phases. Please rewrite this sentence according to the facts.
2 - There appears to be an extra peak with significant weight around 2θ = 43 degrees in all three measured X-ray patterns, which is not seen in the simulated pattern. Can the authors identify possible candidates of these secondary phases?

Response)
At least XRD of PrNi0.8Ge2 indicated the CeNiSi2-type structure as the only phase. Compared to the simulation, the peak around 2θ=43 corresponds to the (002) reflection. To make it clear, we replaced "X-ray analysis of the sample" with "X-ray analysis of PrNi0.8Ge2" in the abstract. Indices are added in the XRD simulation in Fig. 1.

3 - The first paragraph of the Introduction contains a generic description of Kondo lattice physics, which seems irrelevant to the rest of the manuscript. So, I recommend the authors to simply remove those sentences from the introduction. Instead, I recommend the authors to expand the introduction on the reported magnetic properties in RTxGe2, and how that compares with PrNixGe2.

Response)
We have deleted the first paragraph of the Introduction.

4 - Throughout the manuscript, the ferromagnetic ordering temperature stated by the authors has been varying, for instance, 14 K in the abstract but 12 K in the conclusion. Please specify a clear criterion, based on which the ordering temperature has been determined, and keep the value consistent throughout the manuscript.

Response)
Tc is estimated from the result of heat capacity. So, we replaced 14 K with 12 K in abstract.

5 - In Table 1, the lattice parameters for x = 0.8 and x = 0.6 are identical. I believe this is a typo. Please correct it.

Response)
We thank for pointing out our serious mistakes. We carefully recalculated and revised Table 1. We found that the unit cell volume does not decrease as decreasing x. So, the relevant sentence in the abstract and the first paragraph in section 3 has been deleted.

6- Please specify how the lattice parameters are determined from the powder X-ray diffraction patter, for example by Rietveld refinement using which software.

Response)
The lattice parameters a, b, and c are determined by a least-squares approximation from the d-values corresponding to the reflections. We have added an explanation to the first paragraph in section 3. Indices are added in the XRD simulation in Fig. 1.

7 - The EDX result reported by in Section 2 for the three samples appear to vary strongly between the three samples. In particular, the measured composition for x = 0.8 seems to be significantly deficient in Ge. Can the author quantify the error on these composition measurements or provide an explanation as to the strong variation in the measured compositions?

Response)
The composition obtained is an experimental fact. There may be problems with the experimental setup, but we cannot quantify the error. So, we deleted the sentence "These results are closed to the theoretical value..."

8 - For the fitting of magnetisation data to Equation (1), please specify the temperature window for which the fitting has been carried out in each case.

Response)
Fitting was carried out in the temperature range 100 K < T < 180 K. We have added a sentence on page 3, line 4 from the bottom and on page 5, line 12.

---

## Editorial Decision

published